# A Novel Nanomaterial-Based Approach for the Cryopreservation of Individual Sperm Cells Using Addressable Nanoliter Containers

**DOI:** 10.3390/nano15030149

**Published:** 2025-01-21

**Authors:** Bat-Sheva Galmidi, Yana Shafran, Chen Shimon, Adva Aizer, Raoul Orvieto, Naomi Zurgil, Mordechai Deutsch, Zeev Schiffer, Dror Fixler

**Affiliations:** 1The Biophysical Interdisciplinary Jerome Schottenstein Center for the Research and Technology of the Cellome, Physics Department, Bar-Ilan University, Ramat-Gan 5290002, Israel; bat77gal@gmail.com (B.-S.G.); yana.shafran@gmail.com (Y.S.); zurgiln@gmail.com (N.Z.);; 2Department of Gynecology and Fertility, Sheba Medical Center, Tel HaShomer, Ramat-Gan 5262000, Israel; chen.shimon@sheba.health.gov.il (C.S.); adva.aizer@sheba.gov.il (A.A.); raoul.orvieto@sheba.health.gov.il (R.O.); 3Department of Laser Physics, Elbit Systems, Rehovot 7670305, Israel; zeev.schiffer@elbitsystems.com; 4Faculty of Engineering, The Institute of Nanotechnology and Advanced Materials, Bar-Ilan University, Ramat-Gan 5290002, Israel

**Keywords:** cryopreservation, nanoliter, sperm cells, diffusion, IVF

## Abstract

The research and development of a matrix of Addressable Nanoliter Containers (ANLCs) is the focus of this work. ANLCs introduce a novel approach for cryopreserving single sperm cells. A significant increase in sperm cell mortality was observed after cryopreserving nanoliter-scale cell suspensions, attributed to the diffusion of water from the aqueous droplets into the surrounding oil phase. This process elevated the salt concentration within the droplets. A practical solution was devised by saturating the oil with water, significantly reducing the concentration gradient and, consequently, the diffusion. For ANLCs smaller than a few nanoliters, locating individual sperm cells within the containers became highly feasible. Using saturated oil, the survival rate reached 100%. Optical simulations were conducted to evaluate the impact of ANLCs on light scattering, enabling the selection of designs with minimal scattering. The simulations conclusively demonstrated that a cylindrical container with a flat bottom produced the least light scattering. This device was tested under clinical conditions in an in vitro fertilization (IVF) laboratory, revealing its strong potential as a practical tool for housing individual sperm cells. It enables characterization using interferometric indicators and facilitates the selection of sperm cells for IVF.

## 1. Introduction

A wide range of medical conditions, along with various biological and environmental factors, can temporarily or permanently lead to low sperm count, necessitating the freezing and retrieval of only a small number of sperm cells [1]. The use of non-ejaculated sperm in conjunction with intracytoplasmic sperm injection (ICSI) has become a well-established procedure for men with azoospermia. Additionally, surgical techniques have been developed to retrieve spermatozoa from the epididymis and testes of these patients [2]. To minimize the need for repeated surgical interventions, cryopreservation of very few sperm cells—even a single cell—is essential.

Conventional sperm cryopreservation methods are unsuitable for small quantities of sperm cells, such as those retrieved from the epididymis or testes. When individual cells are cryopreserved using standard cryo-tubes, they are undetectable after thawing due to the large volume of the container [3]. Efficient cryopreservation of surgically retrieved spermatozoa reduces the need for repeated surgical procedures and circumvents logistical challenges associated with coordinating oocyte and spermatozoa retrieval. This significantly lowers the risks of complications, such as testicular damage, epididymal fibrosis, testicular atrophy, and deterioration of spermatogenic function [4,5,6]. Various methods for cryopreservation of small numbers of human spermatozoa have been proposed, each offering distinct advantages alongside limitations and drawbacks. Table 1 summarizes some of these methods, along with their main disadvantages.

However, to date, no clinical trials have demonstrated the superiority of one carrier over another, and no fully standardized technique for cryopreservation of a single human spermatozoon has been adopted by in vitro fertilization (IVF) laboratories [3,17,18].

In this manuscript, we propose using nano methods to solve the challenge—a matrix of Addressable Nanoliter Containers.

## 2. Materials and Methods

### 2.1. Materials

Spermatozoa (about 20 samples from anonymous donors) were obtained from the Infertility and IVF Unit of the Chaim Sheba Medical Center, Tel-Hashomer, Israel, from 1 January 2018 to 31 December 2022. Spermatozoa were derived from ejaculated samples that remained redundant after testing or fertilization; we did not use cells collected during surgical procedures.

Quinn’s Advantage Sperm Freezing Medium, (CooperSurgical, Ballerup, Denmark) and oil for tissue culture was purchased from Sage (SAGE In Vitro Fertilization, Målov, Denmark).

### 2.2. Ethics

Sperm was obtained after receiving signed informed consent from each patient to use cells that would have otherwise been discarded. All personal data were fully anonymized to maintain patient privacy, and samples were coded so that patient information could not be reached. This study was approved by the local Institutional Review Board (approval number 0187-23-SMC).

### 2.3. Measurement System

Images were acquired using a motorized Olympus inverted IX81 microscope (Tokyo, Japan), which is equipped with a sub-micron Marzhauser Wetzlar motorized stage type SCAN-IM, with an L step controller (Wetzlar-Steindorf, Germany) and a filter wheel including a fluorescein fluorescence cube containing excitation filters (470–490 nm), dichroic mirrors (505 nm long pass), and emission filters (510–530 nm). The filters were obtained from Chroma Technology Corp. (Brattleboro, VT, USA). A cooled, highly sensitive 14-bit, ORCA II C4742-98 camera (Hamamatsu, Japan) was used for imaging. Olympus Cell^P software version 1.7 (Tokyo, Japan) was used for image analysis.

### 2.4. Freezing and Thawing Sperm Cells

Spermatozoa washing medium and freezing medium were mixed in a 1:1:2 ratio. The Petri dish was placed in liquid nitrogen vapor for 5–10 min and then immersed in liquid nitrogen for the freezing process. For thawing, the Petri dish was transferred to a microscope and incubated at 37 °C.

### 2.5. Developing the Addressable Nanoliter Containers

Efforts to trap a small number of cells in a given volume for various purposes have taken more than a decade, and most are reviewed in the Introduction. Consequently, ANLC arrays have been developed, in which each (about 2.5 nL in volume) acts as an individual isolated ANLC, wherein an individual live sperm cell can be captured and monitored under a microscope noninvasively and in a time-resolved manner. A series of different shapes were investigated including a cylinder with a flat or a concave bottom and an inverted truncated pyramid with different convergence angles. After researching the optical effects that the shapes may exhibit when examining the ANLC and the sperm cell model inside it with an interference microscope, we chose the first option. The ANLC was designed and constructed to accommodate the freezing and thawing conditions required for cryopreservation.

The ANLC array (Figure 1) was fabricated using a Photo Lithographic Patterning technique. A sheet of 175 µm thick borosilicate glass, type D263 (GeSiM, Radeberg, Germany) was spin-coated at 3500 rpm with SU8-5 photoresist, to a thickness of 2–2.5 µm. The ANLC array was patterned on the photoresist by illuminating it through a prefabricated chromium mask. This was followed by thermal annealing at 175 °C for 60 min, resulting in stiff, smooth surfaces of the structured SU8-5 ANLC. Finally, the arrayed glass was sawed into 5 × 5 mm^2^ chips, cleaned (mainly from glass debris) using water jetting, dried with clean compressed air, and kept in antistatic bags until fabrication. Then, the ANLC arrays were glued onto a standard microscope slide with a droplet of NOA81 (Norland Optical Adhesives, Jamesberg, NJ, USA) and cured with UV light for 25 s. The ANLC array was suited for conventional cryopreservation by gluing it to a spoon-like handle, made of plastic or 0.3 mm aluminum foil, and then stored in a standard cryo-tube (Figure 2).

Once the preparation of the ANLC print was completed, it was treated in an electronic plasma system (Diener Electronic GmbH & Co., KG, Ebhausen, Germany). This plasma system is used for surface cleaning, surface activation, surface etching, and surface deposition treatment to give the print hydrophilic characteristics. The print was then sterilized for 2 min with UV light (Model 5000 Flood Light Curing System, Dymax Corp., Torrington, CT, USA) and packaged using a household vacuum machine into individual plastic bags.

Ways to enhance hydrophilic characteristics:Using “cold plasma” (ours heats the prints and works without oxygen);Using more professional vacuum machines and plastic bags;Storing prepared and sealed prints at low temperatures.

An ANLC attached to a plastic handle cannot be treated with plasma because it melts during plasma treatment. However, if the handle is made of 0.3 mm aluminum foil, it can be plasma treated. In the former case, the plastic handle with the attached ANLC array was soaked in alcohol for 24 h. After this, it was washed with water and medium and sealed in cryo-tubes (Eppendorf, Hamburg, Germany) soaked within the medium to be used.

### 2.6. Optical Aspects of Addressable Nanoliter Containers (ANLCs)

The objective of fabricating a nanoliter size container (ANLC) is based on the goal of improving the ability to localize a sperm cell accurately. On the other hand, reducing the ANLC container dimensions might cause increased clutter caused by the effect of scattered light. To demonstrate the usefulness of the suggested ANLCs, we examined the effect using a highly phase-sensitive measurement procedure. Therefore, an interference microscope was used (Figure 3).

In short, an interference microscope is principally designed to enable imaging of a desired interference pattern [19,20]. The pattern is obtained by splitting an incoming monochromatic beam into illumination and reference beams. One passes through the measured object and, hence, acquires a location-dependent optical delay. This optical delay may also be referred to as an optical phase distribution of the light wave field. This field may be imaged on the camera plane by means of an imaging lens. The second beam, which serves as a phase reference beam, is then directed into the (camera) plane, resulting in an interference (typically fringe raster) pattern. This pattern encodes the phase information that can be decoded by means of analyzing algorithms.

The system could be numerically simulated, thus enabling a theoretical investigation of the effect of various micro-well designs on the accuracy of the retrieved optical pass length distribution, as deduced by the interferometric microscope model.

The results obtained by simulation were amazingly similar to the actual results [19,20] in the findings of Shaked (2012) and, therefore, permitted the influence of the ANLC on measurements of interferometer microscope to be examined [21].

### 2.7. The Numerical Model

The numerical simulation model [22] included three main modules as depicted in Figure 3: A interferometer microscopy setup, which comprises wave propagation of a Gaussian beam through various optical components of the device, based on the actual optical system to be investigated; B. a sample model, which includes detailed 3D optical characteristics of the sperm cell and the ANLC; and C. the ANLC, which hosts a sperm cell during the process.

The 3D model of the sperm cell is elaborated based on the geometrical and optical parameters of a typical sperm cell, obtained by means of optical coherence tomography (OCT). The full process of phase extraction is simulated by means of an algorithm fed with synthetic interferometric data.

As described in the data analysis section below, a fixed beam waist was used to illuminate the ANLC with the sperm cell model in it. The waist diameter of the input beam was 60 µm in the sample plane, with a depth-of-field of a few hundreds of microns, which is much larger than the height of the ANLC. The beam wavelength could be simulated to have a range of 0.8 > λ > 0.4 µm. To obtain the numerical value of the spot diameter, a condenser lens was used to focus the beam on the sample, while an imaging lens was used to create a magnified image on the camera plane. A reference beam was directed at the desired angle, typically a few degrees, to interfere with the electric field of the imaging arm at the camera image plane. By modulating the phase of the reference beam, the typical fringe pattern is affected, making it possible to extract the relative spatial phase delay induced by the sample optical density distribution. The result may be presented in the form of a topographic phase map, indicating the combined effect of the size, shape, and depth of the sperm cell under investigation. Alternatively, the optical density could be estimated by geometrical assumptions, avoiding the need for phase modulations.

The sample model was composed of an ANLC with dimensions in the range of 100–150 µm and depth in the range of 90–110 µm. The sperm cell was modeled according to its characteristic optical and physiological parameters, as described by Shaked (2012) [19], with dimensions and a refractive index that comprised a 3D distribution.

The simulation system comprising the features described was adjusted to mimic several experimental situations, all of which could be compared to results reported by Shaked (2012) [19].

### 2.8. Data Analysis

During the measurement process, the camera plane is illuminated by two beams. The first is the beam coming from the sample branch, and the second is the reference beam. The interference between these creates fringes of the form presented in Figure 4A. As the phase of the reference beam is modulated, the pattern of the fringes evolves, thus enabling extraction of the relative phase delays at different points in the sample. These phase delays are caused by the optical density of the specimen, which depends on, among other factors, the size and contents of the sperm cell head.

As seen above, all phase data can be recovered very accurately by using the above-mentioned set of fringe patterns. The main limitation in the precision of the phase extraction process is dictated by diffraction due to the finite wavelength used and to the apertures along the collection optical path.

The main effect of the ANLC could be on the wave front aberrations and deviations induced by the irregular shape of its walls and bottom. To check the effect of these aberrations, we examined the effects of four ANLC designs on the quality of the phase shift extracted by the interference microscope (Figure 5). These four designs were as follows: Figure 5A. pyramidal walls with the tip having dimensions of about 10 µm, Figure 5C. cylinder with a concave bottom, Figure 5E. ANLC in the shape of a truncated pyramid with the tip having typical dimensions of about 80 µm, and Figure 5G. cylinder with flat bottom. Figure 5 presents these four types of ANLCs and the corresponding images of sperm cell phase distribution expected to be extracted by the interferometer microscope. In Figure 5B, a considerable halo is seen around the area of interest when compared to the actual or original phase distribution as in Figure 5F,H. Although with the spherical-bottom ANLC (Figure 5C), the halo is less pronounced in Figure 5D, but it still has a blurring effect on the extracted image when compared to Figure 5F and Figure 5H (Figure 5E) and (Figure 5G) the cylindrical ANLC, where the retrieved phase distribution does not have these significant distortions.

Figure 6 shows numerical examples of these four ANLC designs. The geometrical cross-section of the micro-well optical paths of the designs are presented in Figure 6A. Figure 6B presents the distributions of the cross-sections corresponding to the retrieved phases along the x-axis. The retrieved phase cross-section of the flat design matches the actual values, but the one obtained for the hemi-spherical micro-well has significant deviations and distortions with respect to the actual set of values.

### 2.9. Physical Summary of Tolerance

The simulation system presented in this work was designed to study the expected phase extraction of an interference microscope system in the four ANLC designs carrying a model of a typical sperm cell. The microscope system’s performance in the different ANLC clearly shows blurring and distortions caused by the optical properties (i.e., shape, size, etc.) of the ANLC designs. The flatness of the ANLC is a critical parameter, where the area illuminated by the beam was much larger than the sperm cell itself. However, the best performance was found with a flat-bottom, cylindrical ANLC, which was consequently used in this study.

### 2.10. Software to Analyze Sperm Cell Mobility

To obtain the best statistical information for dozens or hundreds of motile sperm cells in a droplet, novel software (SMA: Sperm-Mobility- Analyst, version 0.19) was developed to analyze sperm cell mobility, based on a convolutional neural network. The software tracks a video (see Appendix A) clip taken under a microscope, learns to identify sperm cells, and tracks their route. It outputs the percentage of motile cells and their average velocity. The software was designed to provide highly accurate and detailed insights into sperm function. It incorporates several of the following key features:(1)Precise Measurements: Calculations are performed in micrometers, with frame rate and pixel size factored in, for high-resolution analysis.(2)AI-Powered Object Detection: A pre-trained convolutional neural network model enables accurate detection of individual sperm cells.(3)Motion Tracking: The software tracks sperm movement paths across sequential image frames and identifies sperm cells and head–tail orientation.(4)Data Filtering: The software removes immotile and abnormally motile sperm to enhance data quality and focus on viable cells.(5)Involuntary Movement Analysis: Involuntary movements deviating significantly from the sperm’s axis are identified and quantified. The external (collective) motion measured is subtracted from the total velocity of the cell.(6)The times when a cell enters and leaves the field of view are recorded and evaluated accordingly.(7)Robust Metrics: The software calculates the percentage of progressively motile sperm and average speeds, providing key indicators of sperm quality.

The software is designed to provide the following information:

Identify sperm cell and head–tail orientation;

-Detect movement towards the head, ignoring movement that is not forward;-Detect external flow motion and subtract it from the total velocity of the cell;-Recognize the point in time when objects enter and leave the image, as well as focus on and calculate only the time in which the image is displayed in the clip; thus, the clip will be weighted according to time.

## 3. Results

### 3.1. Storage Efficiency of Liquid Within ANLCs During Freezing

Preliminary experiments show that fluid does not exit the vessel during the freeze–thaw cycle. Figure 7A shows fluorescent images of the vessel structure within a 5 μM fluorescein drop prior to the freezing–thawing cycle, and Figure 7B shows images following the freezing–thawing cycle after partial removal with blotting paper.

### 3.2. Cryopreservation of Molt 4 Cells

Fluorescein diacetate (FDA) and propidium iodide (PI) (Sigma-Aldritch) fluorescent dyes were added to the Molt 4 cell suspension. Then, the suspension was loaded onto the ANLC array and into a standard cryo-tube (control). The ANLC array was carefully washed using a pipette with Quinn’s Advantage Sperm Washing Medium (Al-Rad Medical, Ltd., Ness Tziona, Israel), and a bright field image of the same region was taken before (Figure 8A) and after (Figure 8B) washing. It is evident that cells that are lodged between ANLCs can be easily washed out, while cells within the ANLC retain their position. The fluorescent image (Figure 8C), taken after thawing, demonstrates that most cells inside the ANLC are alive. The green fluorescing spots are FDA-positive (live cells), and the red spots are PI-positive (dead cells). Similar experiments were performed using vessels of various diameters, with volumes of 5 µL, all showing similar results.

### 3.3. Cryopreservation of Individual Sperm Cells

Figure 9 illustrates the step-by-step process of using the ANLC device from sample loading to sperm identification and isolation. Sperm cells were collected using a micro-manipulator pipette. The pipette, via the micromanipulator, was brought into the proximity of the sperm cells (suspended in a drop of medium, covered by a layer of oil (Figure 9A)), after which cells were collected by gentle pumping. Then, the collected sperm cells were released into a cryo-preservation medium inside vessels with varying diameters, covered by oil (Figure 9B,C). The sperm cell suspension was also loaded into a standard cryo-tube for the control experiment.

The tubes were sealed and placed in liquid nitrogen vapor for 10 min, then immersed in liquid nitrogen at −196 °C. For thawing, the tube was heated for 3 min at 37 °C, under oil.

Results with sperm cells show that (a) following the freezing–thawing procedure, about 90% of the cells remained in the original vessels; (b) about 80% of the motile cells were in the μL volume vessels, which was even more than in the control experiment; and (c) as opposed to (b), none of the sperm cells in the ANLC survived.

### 3.4. Saturating the Oil with Water

To overcome this shortcoming and to allow for cryopreservation of individual sperm cells, we propose a simple, user-friendly approach of replacing oil with water-saturated oil. This has been found to significantly augment the ability of sperm cells to survive the harsh freeze–thaw cycle. To obtain saturated oil with water, oil was mixed with water using a magnetic stirrer for an hour at a temperature of 37–40 °C. Then, the oil layer was carefully extracted and used for cryopreservation. The exact nature of the saturated oil is described in detail in our previous work [23,24]. A significant improvement in the percent motility after thawing and in velocity after soaking in the freezing medium was observed.

### 3.5. A Clinical Study in the IVF Department at Sheba Hospital

After extensive examination of several devices, the spoon-like carrier was found to be most convenient for freezing sperm cells, and it was used to test the beneficial effects of saturated oil on in-device cell survival under clinical conditions. The ANLC was inserted into a low-pressure plasma system (Femto, Diener Electronic, Ebhausen, Germany) so the ANLC would become hydrophilic. This process is necessary to increase adhesion forces and to allow the aqueous solution to sink to the bottom of the ANLC. Freezing was performed as usual, except that this time the saturated oil was used. The thawing procedure was performed as usual.

A distinct improvement was observed in cell survival after thawing. Because these were single cells, the statistics are inaccurate, but the IVF lab reported 50% survival using saturated oil, as compared to no survival using regular oil.

## 4. Discussion

This study proposed and evaluated nanomaterial methods and procedures for the cryopreservation of single sperm cells within 1.5 ANLC. Minimizing the dimensions of the ANLC enhances the ability to locate the sperm cell; however, optical interference may affect the wavefront of the incoming laser beam and the light scattered by the immobilized sperm cell. This aspect was thoroughly investigated through computer simulations of an interference microscope. The results indicated that the ANLC design that minimizes light interference is a cylindrical container with a flat bottom.

Additionally, the fact that survival of sperm cells following a freezing–thawing cycle in relatively large volumes was found to be much larger than that obtained within the ANLC compelled us to examine, more precisely, the dependency of sperm cell vitality on the freeze–thaw volume. We found that in droplet volumes below a few nanoliters, the greater the decrease in droplet volume, the more rapid the increase in solute concentration within, tending to infinity. This evidently prevents survival of sperm cells within these droplets. Covering the watery droplet with oil did not prevent the transfer of water molecules through it and hence did not prevent the elevation of solute concentration within the droplet. All of this is detailed in a work that has been submitted for publication and has not yet been published.

A different study investigated the issue of a nanoliter droplet covered by oil and its unique characteristics [23], which is part of the method used for cryopreservation of individual sperm cells in an ANLC. An in-depth experimental and analytical study [23,24] established that this phenomenon is caused by water diffusing from the droplet before freezing into the surrounding covering oil media, which increases the solute concentration within the droplet. As a result, the concentration of salts in the vicinity of the sperm cell increases, and the cell dehydrates. The poor physiological condition of the cell makes it difficult for it to survive the freezing and thawing process. The strong reciprocal dependency of the rate of change in droplet solute concentration (C), upon the water volume (V), ∝, teaches us that with ANLC, fewer cells will survive compared with microliter volumes, due to the more rapid increase in solute concentration. This study concluded that the proposed ANLC, combined with water-saturated oil, is an effective and user-friendly approach for cryopreserving individual sperm cells. The ANLC matrix can be integrated into various components, such as microscope slides, cover slips, or Petri dishes. In this work, a spoon-like carrier with a tweezer-like handle was proposed for ease of handling. For optimal results, the water-in-oil mixture should be prepared in advance, and the sperm cells should be injected into the ANLC under the oil layer. The device can then be placed into a standard freezing tube, allowing the freezing and thawing processes to remain consistent with conventional procedures.

Dedicated software designed for the container matrix can easily and rapidly scan numerous cells in a short period. Moreover, the ANLC device is compatible with any scanning software that works with the microscope being used, as each container is assigned a precise address and specific coordinates. This feature facilitates not only efficient searching and identification of sperm cells but also allows for the potential scalability of our method in clinical applications.

We believe that the ability to cryopreserve individual sperm cells within preselected ANLCs could pave the way for new advancements in IVF treatment.

## Figures and Tables

**Figure 1 nanomaterials-15-00149-f001:**
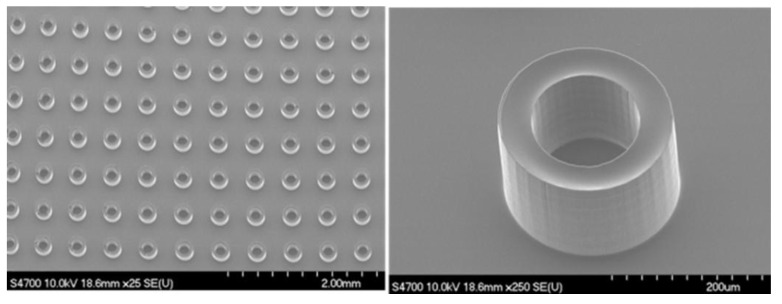
ANLC (addressable nanoliter container) array under microscope. (**Left**) Part of the SU8-5, 5 × 5 mm^2^ chip. Each compartment acts as an individual, isolated reaction ANLC, in which individual live cells can be captured. (**Right**) Side view of a single ANLC (diameter: 250 μm, height: 250 μm⟹ volume: 1.2 × 10^−9^ L).

**Figure 2 nanomaterials-15-00149-f002:**
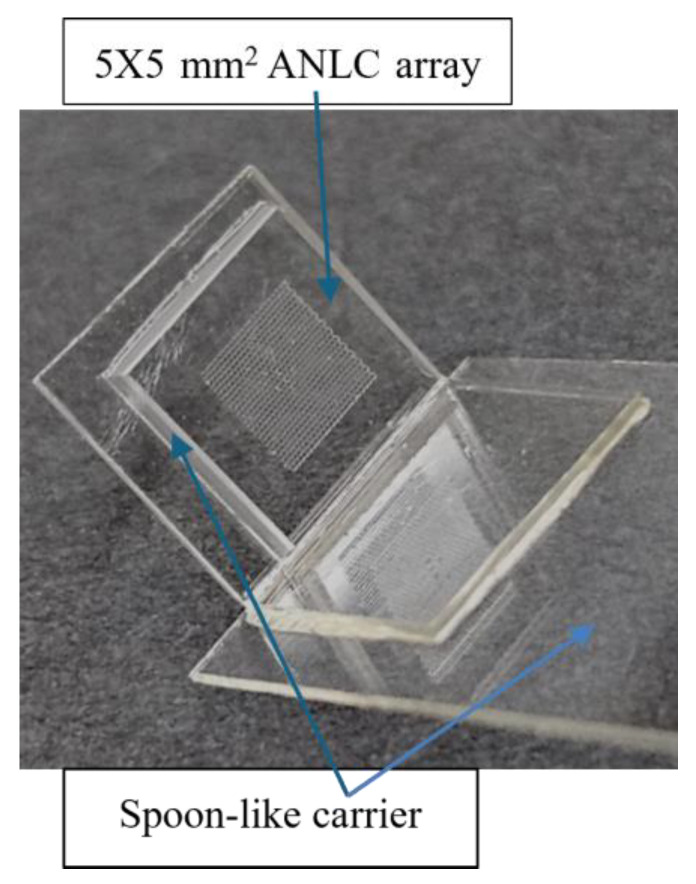
A standard cryo-tube used here for storing the ANLC (addressable nanoliter container). The ANLC array was suited for conventional cryo-preservation by gluing it to a spoon-like carrier and storing it in a standard cryo-tube.

**Figure 3 nanomaterials-15-00149-f003:**
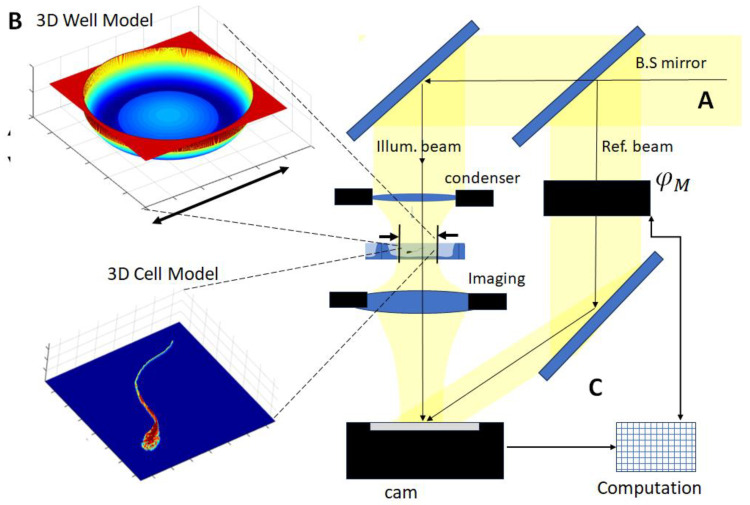
Components of the simulation model. (**A**) Interferometric microscope: The monochromatic light beam is split into reference and imaging illumination arms, along which a condensing lens is used to create the desired location on the specimen plane (φM stands for modulator). As light passes through the specimen, its phase is shifted according to the distribution of the optical path length. The outgoing beam is transmitted into the camera plane (cam) by means of an imaging lens. The phase-shifted reference beam (**C**) is directed into the same camera plane where an interference pattern is created. (**B**) Model of the ANLC with its geometry, which includes shape distortions and relevant optical indices of refraction, and a 3D model of the sperm cell, which is to be analyzed by the interferometric microscope.

**Figure 4 nanomaterials-15-00149-f004:**
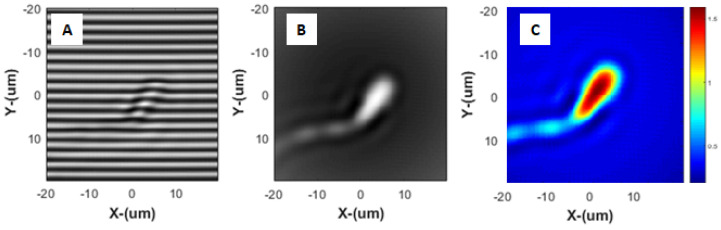
Analysis stages of interferometric data. Distances are measured in μ(u)m. (**A**) The interference pattern obtained on the camera plane is composed of fringes where the spatial separation depends on the illumination wavelength and angle of incidence given to the reference beam. (**B**) Small deviations from the interference fringes raster are attributed to phase shifts imposed on the beam by the specimen under investigation, where the ideal phase distribution is directly related to the actual original optical path length phase distribution on the camera. Using an algorithmic process, the retrieved, calculated, and enhanced phase optical path distribution shown in (**C**) can be obtained.

**Figure 5 nanomaterials-15-00149-f005:**
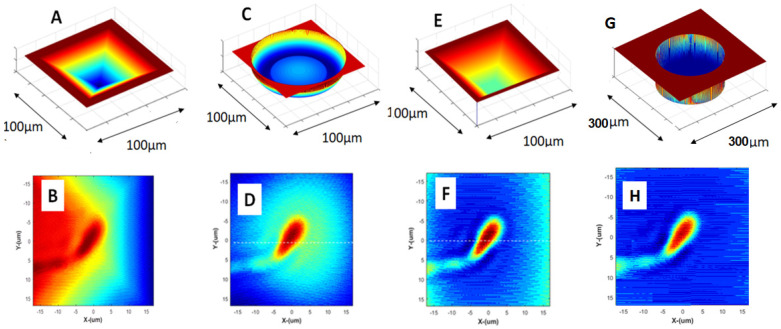
Aberrations related to the different ANLC designs on the retrieved phase accuracy for four types of ANLCs. (**A**) Pyramid, (**B)** its corresponding retrieved phase distribution interference microscope image (**C**) cylinder with a concave bottom result in (**D**), where the retrieved phase distribution has a significant halo around the area of interest in the specimen. (**E**) The shape of a truncated-pyramid ANLC results in (**F**). (**F**) The retrieved phase distribution has no significant distortions when compared to the ideal phase image as is depicted in the (**G**) cylinder with a flat bottom and (**H**).

**Figure 6 nanomaterials-15-00149-f006:**
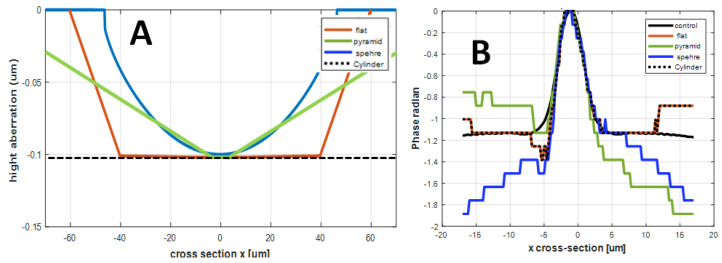
Cross-sections of the 4 ANLC designs. (**A**) Pyramidal (green), cylinder with a concave bottom (blue), truncated inverted pyramid (orange), and a cylinder (dashed black line). (**B**) Corresponding retrieved phase cross-sections.

**Figure 7 nanomaterials-15-00149-f007:**
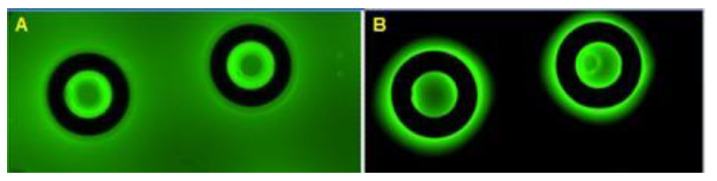
Fluorescein-based preliminary freezing–thawing cycle experiment. (**A**) Fluorescent image of vessel structure with a 5 µM fluorescein drop prior to the freeze–thaw cycle and (**B**) following the freeze–thaw cycle after partial removal with blotting paper.

**Figure 8 nanomaterials-15-00149-f008:**
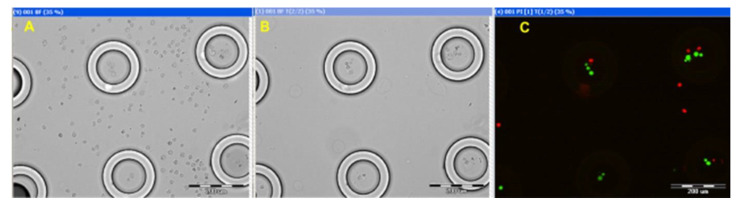
Cryopreservation of individual Molt 4 cells within ANLCs. (**A**) A bright field image was taken after the suspension was loaded onto the ANLC array and (**B**) ANLC array after careful washing. (**C**) Viability test of the cells following the freeze–thaw cycle. The green spots are FDA-positive (live Molt 4 cells), and the red spots are PI-positive (dead Molt 4 cells).

**Figure 9 nanomaterials-15-00149-f009:**
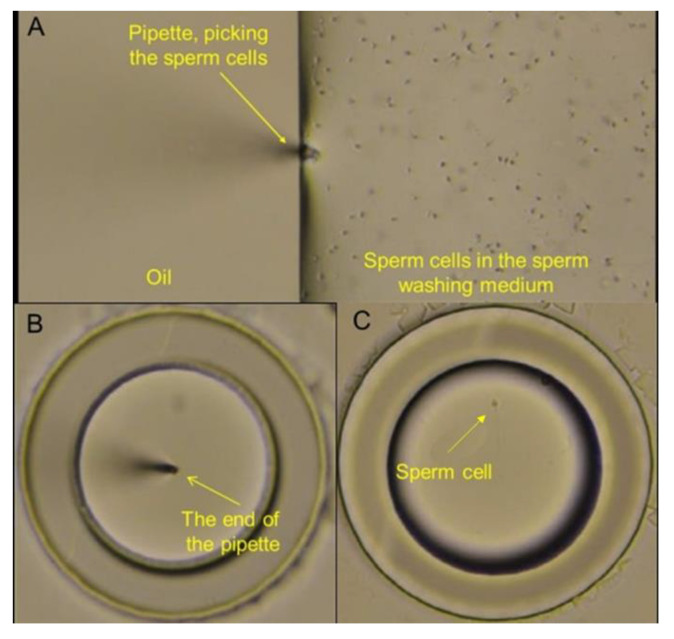
Micro-manipulation technology for cryo-preservation of individual sperm cells in the ANLC. (**A**) Sperm cell collection using a pipette which reaches the sperm-washing medium drop through a covering oil layer. (**B**) Release of the collected sperm cell into cryo-preservation medium inside the ANLC covered by oil (with an air bubble trapped inside). (**C**) Single sperm cell inside a single vessel in cryo-preservation medium (see Appendix A).

**Table 1 nanomaterials-15-00149-t001:** Advanced methods for cryopreservation small numbers of sperm cells and their main disadvantages.

Cryopreservation Techniques	Main Disadvantages
Microdroplets [3,7]	Risk of cross-contamination; shape and size of dishes make it difficult to handle and store in conventional freezers and liquid nitrogen tanks
ICSI pipette [8,9]	Not practical for long-term storage; fragility of ICSI pipettes; risk of cross-contamination
Empty zona pellucida [10,11,12]	Risk of biological contamination
Volvox globator spheres [13]	Exposure to genetic material from the algae; constant source of algae
Alginate beads [14]	Decrease sperm motility with encapsulation
Agarose microspheres [15]	Clinical value of this approach not evaluated
Cryoloop [16]	Open system; risk of cross-contamination

## Data Availability

All data generated or analyzed during this study are included in the manuscript.

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
