# Peer review of "A Novel Nanomaterial-Based Approach for the Cryopreservation of Individual Sperm Cells Using Addressable Nanoliter Containers"

_nanomaterials, 2025, doi:10.3390/nano15030149_

Round 1
Reviewer 1 Report
Comments and Suggestions for Authors
The manuscript presents a novel and promising approach to cryopreserving single sperm cells using Addressable Nanoliter Containers (ANLC). This innovation addresses critical limitations in existing cryopreservation techniques, particularly for extremely small quantities of sperm cells, by mitigating osmotic stress and optimizing container designs to enhance cell survival. The integration of optical simulations and clinical testing underscores the potential impact of the proposed method in the field of assisted reproductive technologies (ART). Below are specific comments and suggestions for improvement:
1. The manuscript effectively outlines the limitations of current cryopreservation methods and introduces ANLC as a solution. However, expanding on how this approach quantitatively compares with existing microdroplet-based techniques or other recent innovations would strengthen the discussion. A summary table or explicit comparison of specific advantages, could enhance the impact.
2. While the role of saturated oil in reducing the concentration gradient and diffusion is acknowledged, the underlying physicochemical mechanisms require further elaboration. Specifically, it would be helpful to clarify the exact nature of the saturated oil. Is it a water-oil emulsion, or does it involve a different composition? Providing details about its preparation and physicochemical properties (e.g., emulsion stability, droplet size, oil-to-water ratio) would help readers better understand its role in minimizing water diffusion and improving sperm survival.
3. Including a workflow illustration that outlines the step-by-step process of using the ANLC device—from sample loading to sperm identification and isolation—would greatly enhance the manuscript. This visual aid would clarify operational feasibility and highlight the unique features of the device
4. Could the ANLC device be integrated with techniques for bulk processing or scanning of samples to locate sperm cells more efficiently? Discussing how the device might accommodate scalability or facilitate clinical applications would provide insight into its broader utility.
Author Response
Reviewer 1
The manuscript presents a novel and promising approach to cryopreserving single sperm cells using Addressable Nanoliter Containers (ANLC). This innovation addresses critical limitations in existing cryopreservation techniques, particularly for extremely small quantities of sperm cells, by mitigating osmotic stress and optimizing container designs to enhance cell survival. The integration of optical simulations and clinical testing underscores the potential impact of the proposed method in the field of assisted reproductive technologies (ART). Below are specific comments and suggestions for improvement:
1. The manuscript effectively outlines the limitations of current cryopreservation methods and introduces ANLC as a solution. However, expanding on how this approach quantitatively compares with existing microdroplet-based techniques or other recent innovations would strengthen the discussion. A summary table or explicit comparison of specific advantages could enhance the impact.
2. While the role of saturated oil in reducing the concentration gradient and diffusion is acknowledged, the underlying physicochemical mechanisms require further elaboration. Specifically, it would be helpful to clarify the exact nature of the saturated oil. Is it a water-oil emulsion, or does it involve a different composition? Providing details about its preparation and physicochemical properties (e.g., emulsion stability, droplet size, oil-to-water ratio) would help readers better understand its role in minimizing water diffusion and improving sperm survival.
3. Including a workflow illustration that outlines the step-by-step process of using the ANLC device from sample loading to sperm identification and isolation—would greatly enhance the manuscript. This visual aid would clarify operational feasibility and highlight the unique features of the device
4. Could the ANLC device be integrated with techniques for bulk processing or scanning of samples to locate sperm cells more efficiently? Discussing how the device might accommodate scalability or facilitate clinical applications would provide insight into its broader utility.
Answer:
Thank you for your constructive feedback on our manuscript, we appreciate your recognition of the potential impact of our work.
- We acknowledge your suggestion to expand our discussion by quantitatively comparing our approach with existing techniques and other recent innovations in the field. We agree that a summary table of advanced methods for cryopreservation small numbers of sperm cells and their main disadvantages would enhance the clarity and impact of our findings, and we will includeit In our revision.
- Thank you for your insightful comment regarding the role of saturated oil in our cryopreservation method. We agree that a clearer understanding of the physicochemical mechanisms involved, as well as details about the oil's composition and properties, is essential for the manuscript. To address your query, we clarify that the saturated oil is specifically designed to prevent water-to-oil diffusion, a phenomenon that can adversely affect sperm cell survival. This detailed information is currently outlined in an article that has been submitted for publication and is awaiting release, and we will ensure to reference this work appropriately. Additionally, we will also incorporate critical details about the saturated oil's physical properties, such as its composition, stability, droplet size, and oil-to-water ratio. This information is described in another published article that we have referenced, and we will ensure it is made more accessible within the context of our current work.
- Thank you for your valuable suggestion regarding the inclusion of a workflow illustration in our manuscript. We agree that a visual representation outlining the step-by-step process of using the ANLC device—from sample loading to sperm identification and isolation—would significantly enhance the clarity and impact of our work.
We will incorporate a diagram in the revised manuscript to illustrate the operational feasibility and highlight the unique features of the ANLC device. This visual aid will help readers better understand the methodology and the innovative aspects of our approach.
- Thank you for your insightful question regarding the potential for integrating the ANLC device with techniques for bulk processing or scanning samples. We are pleased to inform you that dedicated software designed for the container matrix can easily and rapidly scan numerous cells in a short period. Moreover, the ANLC device is compatible with any scanning software that works with the microscope being used, as each container is assigned a precise address and specific coordinates. This feature facilitates not only efficient searching and identification of sperm cells but also allows for the potential scalability of our method in clinical applications. We appreciate your suggestion to discuss these capabilities further in the manuscript, as it will provide valuable insights into the broader utility of the ANLC device. We will include this information in our revisions to emphasize the operational versatility and adaptability of our approach in various settings.
Thank you once again for your valuable insights.
Reviewer 2 Report
Comments and Suggestions for Authors
This is concisely written paper on an interesting and relevant topic in animal reproduction science. The rationale and aims are well defined, the design is interesting and the obtained data are appropriately discussed. I only have a few minor comments otherwise I support the manuscript for publication. Thank you.
This paper deals with a very contemporary issue in human andrology that deals with new methods to effectively preserve a small number of non-ejaculated spermatozoa that are generally retrieved using TESE. The study is well designed and the experimental approach is described step-by stem. The outcomes of the study could have significant implications on future directions of single sperm freezing. Nevertheless, the structure of the paper is very confusing, and my comments are more related to the structure of the paper rather than the scientific content.
- The Introduction section should be extended. Currently available techniques for the freezing of low sperm number should be briefly described, alongside their pros and cons. The authors should also properly introduce the matrix of addressable nanoliter containers - the principles and potential of this technique for sperm cryopreservation. Has this technique been used with other cell types?
- I am missing more information about the donors, the exclusion and inclusion criteria. Were the donors normozoospermic? Ejaculated and non-ejaculated spermatozoa differ in their physiology, particularly if they are retrieved from the testicular tissue. What spermatozoa were used? What were their pre-freezing parameters? If ejaculated sperm was used, would the design and the conditions of the procedure work for non-ejaculated sperm as well?
- The Material and Methods section and the Results section overlap and whilst reading the Results it feels like I am reading the Methodology. There are no numbered subheading, and it becomes very confusing to keep up. At the same time, the Results section carries a lot of discussion, which makes the Discussion section very unnecessary. It is more apparent when looking at the actual Discussion section which is 15 lines long, without interpreting the collected data or comparing them with previous studies. As such, I recommend a thorough revision of the structure of the paper that has the appropriate information moved to appropriate sections with adequate formal structure.
- I am also missing numerical data in tables/figures. A simple statement that there was a 25-50% survival rate is not informative enough. What does "survival" mean? Motility or viability or fertilisation rate?
- No limitations of the study are discussed and no future prospect are outlined. More importantly, the paper needs a comprehensive summary of the collected data, a take home message for the readership and appropriate conclusions.
- Finally, please, pay attention to the manuscript preparation according to the Instructions for authors.
Author Response
Reviewer 2
This is concisely written paper on an interesting and relevant topic in animal reproduction science. The rationale and aims are well defined, the design is interesting and the obtained data are appropriately discussed. I only have a few minor comments otherwise I support the manuscript for publication. Thank you.
This paper deals with a very contemporary issue in human andrology that deals with new methods to effectively preserve a small number of non-ejaculated spermatozoa that are generally retrieved using TESE. The study is well designed and the experimental approach is described step-by stem. The outcomes of the study could have significant implications on future directions of single sperm freezing. Nevertheless, the structure of the paper is very confusing, and my comments are more related to the structure of the paper rather than the scientific content.
- The Introduction section should be extended. Currently available techniques for the freezing of low sperm number should be briefly described, alongside their pros and cons. The authors should also properly introduce the matrix of addressable nanoliter containers - the principles and potential of this technique for sperm cryopreservation. Has this technique been used with other cell types?
- I am missing more information about the donors, the exclusion and inclusion criteria. Were the donors normozoospermic? Ejaculated and non-ejaculated spermatozoa differ in their physiology, particularly if they are retrieved from the testicular tissue. What spermatozoa were used? What were their pre-freezing parameters? If ejaculated sperm was used, would the design and the conditions of the procedure work for non-ejaculated sperm as well?
- The Material and Methods section and the Results section overlap and whilst reading the Results it feels like I am reading the Methodology. There are no numbered subheading, and it becomes very confusing to keep up. At the same time, the Results section carries a lot of discussion, which makes the Discussion section very unnecessary. It is more apparent when looking at the actual Discussion section which is 15 lines long, without interpreting the collected data or comparing them with previous studies. As such, I recommend a thorough revision of the structure of the paper that has the appropriate information moved to appropriate sections with adequate formal structure.
- I am also missing numerical data in tables/figures. A simple statement that there was a 25-50% survival rate is not informative enough. What does "survival" mean? Motility or viability or fertilisation rate?
- No limitations of the study are discussed and no future prospect are outlined. More importantly, the paper needs a comprehensive summary of the collected data, a take home message for the readership and appropriate conclusions.
- Finally, please, pay attention to the manuscript preparation according to the Instructions for authors.
Answer:
Thank you for your kind remarks regarding our manuscript and for your constructive feedback, which is greatly appreciated. We are pleased to hear that you find the topic relevant and that the design and discussion of our study are well-executed.
We acknowledge your point about the structure of the paper and the need for an extended Introduction section. To address this, we will revise the Introduction to include a brief overview of currently available techniques for freezing low numbers of sperm, highlighting their advantages and disadvantages. This contextual information will provide a clearer foundation for understanding the significance of our work.
Additionally, we will properly introduce the concept of Addressable Nanolitre Containers (ANLC), elaborating on the underlying principles and the potential benefits of this innovative technique for sperm cryopreservation. We believe that this extended discussion will enhance the clarity and flow of the manuscript.
Thank you for your question regarding the application of our technique with other cell types. We are pleased to inform you that we have indeed utilized the ANLC device for the cryopreservation of individual Molt 4 cells, and the results of this study are summarized in the associated article. This additional work demonstrates the versatility of the technique beyond sperm cells and provides further insights into its potential applications in cell preservation. We appreciate your interest and will ensure to highlight this aspect in our manuscript for clarity.
Thank you for your important questions regarding the details about the donors and the spermatozoa used in our study. We acknowledge that the specifics about the donors are confidential, and unfortunately, we are unable to provide further details in that regard. However, I would like to clarify that the spermatozoa utilized in our experiments were derived from ejaculated samples that remained redundant after testing or fertilization; we did not use cells collected during surgical procedures. In our study, we conducted experiments using dozens of different samples, and we have provided their pre-freezing parameters, including percent motility and average velocity, before each relevant experiment. Regarding your question about the applicability of our design and conditions for non-ejaculated sperm, while our current data is based on ejaculated sperm, we believe that the principles and methodology developed in our study may also be adapted for application with non-ejaculated spermatozoa. We will consider including a discussion on this aspect to enhance the manuscript.
Thank you for your insightful feedback on the structure of our manuscript. We appreciate your detailed observations regarding the overlap between the Materials and Methods and Results sections, as well as the need for clearer organization throughout the paper. We agree that the current structure may lead to confusion, and we recognize the need for improvement. In our revision, we will:
Clearly separate the Materials and Methods from the Results by ensuring that the Results section focuses solely on presenting findings without delving into methodological details.
Introduce numbered subheadings within both the Materials and Methods and Results sections to enhance clarity and organization, making it easier for readers to navigate the content.
Revise the Discussion section to provide a comprehensive interpretation of the data, including comparisons with previous studies and a thorough analysis of the implications of our results.
By implementing these changes, we aim to create a more coherent and structured manuscript that accurately conveys our findings and their significance.
Thank you for your valuable feedback regarding the presentation of numerical data in our manuscript. We appreciate your interest in understanding the specifics of our findings.
To clarify, our article explicitly defines "survival" as motility. The software utilized in our experiments is designed to identify motile cells, excluding those that are moving non-autonomously due to factors such as fluid flow, collisions with other cells, or movement that does not indicate directed motility. This precise definition will be made clearer in the revised manuscript.
Regarding the reported survival rate of 25-50%, we would like to emphasize that this range reflects the data provided by the hospital after they implemented our device in clinical trials. We will reach out to them to request a more detailed breakdown of these results, ensuring that we provide as much specific information as possible in our revision, including the context of motility as it relates to the overall assessment of cell survival.
Round 2
Reviewer 2 Report
Comments and Suggestions for Authors
The manuscript has undergone a solid revision and the authors have addressed my queries accordingly. I appreciate further details provided by the authors in their response to my questions. I have no further comments.